# Transesterification Using Ultrasonic Spray of Triolein Containing CaO Particles into Methanol Vapor in a 3-Phase Reactor

**Ravisut Vitidsant \*, Satoshi Kodama and Hidetoshi Sekiguchi**

Department of Chemical Science and Engineering, School of Materials and Chemical Technology, Tokyo Institute of Technology, Meguro-ku, Tokyo 152-8550, Japan; skodama@chemeng.titech.ac.jp (S.K.); hsekiguc@chemeng.titech.ac.jp (H.S.)
\* Correspondence: vitidsant.r.aa@m.titech.ac.jp

**Abstract:** Ultrasonic spraying was used in a three-phase reactor to produce small droplets of triolein mixed with CaO as a solid catalyst at temperatures above the boiling point of methanol for enhancement of the transesterification of triolein. Droplets fell in the methanol countercurrent flow and were collected at the bottom of the reactor, followed by circulation to the ultrasonic spray system. The experimental parameters included triolein flow rates of 2.5–9.0 mL/min, reaction temperatures of 70–100 °C, and catalyst contents of 1.0–7.0 wt%. The methanol feed rate was set to be constant. The results suggested that the enhancement was successful after using the three-phase reactor by generating a high contact surface area for the droplets, which was a key factor for determining the performance. Comparing the results with conventional transesterification in the liquid phase using the same CaO at 60 °C, the three-phase reactor produced a methyl ester yield 2–5% higher during the 60 min trial period. However, the yield became lower after 60 min because the mass transfer of methanol to the droplets was limited. The transesterification kinetics were estimated based on the experimental data—assuming a first-order reaction—and the results indicated a range of the rate constant, an apparent activation energy, and a pre-exponential factor of $1.21$–$3.70 \times 10^{-2}$ min$^{-1}$, 36.1 kJ mol$^{-1}$, and 64.9 min$^{-1}$, respectively, suggesting that the three-phase reactor was effective for fast transesterification at the initial stage.

**Keywords:** ultrasonic spraying; three-phase reactor; triolein; transesterification; CaO; methanol vapor

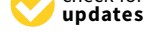

## 1. Introduction

Modern society is currently consuming a great deal of energy. Fossil fuels are one of the types of fuels that have long been used by humans. The enormous consumption of fossil fuels, however, causes a major problem: global warming. Biodiesel has the potential to be one of the clean, sustainable energy sources for human civilization, and it has proven to be an attractive substitute because of its renewability and combustion performance being nearly similar to conventional diesel oil. Generally, biodiesel is produced by the transesterification of vegetable oil, algal oil, or animal fat [1] by using a homogenous catalyst such as sodium hydroxide. The disadvantage of the homogenous process is that the catalyst remains with the product after the reaction. A washing stage is required to purify the product before usage. Also, wastewater from the washing cannot be directly released into the environment. This problem could be solved by changing the homogenous catalysts to heterogeneous catalysts. Alkaline earth metal oxide is considered a candidate because the strength of the basic site is related to the electronegativity of the conjugated metal cation. The basic strength of the data from CO$_2$-TPD (Temperature Programmed Desorption) is in the sequence of MgO < CaO < SrO [2]. The use of CaO [3] as a solid catalyst leads to an economic advantage, due to the ease of recycling the catalyst [4,5] for

simultaneous transesterification and its environmental friendliness. However, because of the heterogeneously catalyzed transesterification that consists of three phases (oil–catalyst–methanol), the reaction rate is quite slow for the mass transfer among the phases. The application of CaO for the transesterification is thus restricted due to the requirement of a long reaction time and the high molar ratio of methanol to oil [6,7].

One way to promote the reaction is to increase the reaction temperature [8]. The typical reaction temperature in the traditional process is less than 60 °C, as the methanol boiling point is 64.7 °C. Methanol evaporates when the temperature exceeds the boiling point, and hence a three-phase reactor was designed where the methanol vapor could react with the liquid triolein and CaO upon heating the reactor over the boiling point of methanol. An ultrasonic spray device was equipped to produce small oil droplets, including CaO. The methanol vapor flowed upward in the countercurrent to the triolein droplets in the reactor. The surface area of the droplets was enlarged, owing to the ultrasonic spraying, which was expected to enhance the mass transfer of the methanol vapor as well as the transesterification reaction. The effectiveness of the proposed reactor was evaluated.

## 2. Materials and Methods

### 2.1. Catalyst Preparation

The CaO catalyst was prepared by the calcination of $CaCO_3$ (Wako Pure Chemical) [9], which was first loaded into a ceramic crucible and then put into the furnace for calcination at 800 °C for 2 h. The calcinated CaO was kept in a desiccator before the experiments, which had a Brunauer–Emmett–Teller (BET) surface area of 12 $m^2$/g.

### 2.2. Transesterification Reaction in the Three-Phase Reactor

Triolein was used as the representative of vegetable oils for the reactant. Transesterification of triolein by a methanol vapor with CaO was carried out in the three-phase reactor, as shown in Figure 1. The experiments were conducted in a chemical fume hood for safety. A mixture of 100 mL of triolein (Cica-Reagent) with a specific weight percentage of CaO was loaded in the reactor before starting. The volume of the reactor was 2 L. The glass reactor was wrapped with a ribbon heater to keep the reactor temperature constant. The reactor was heated to the desired temperature prior to the reaction. A magnetic stirrer was placed at the bottom of the reactor to mix the CaO and triolein well. The peristaltic pump brought the mixture of triolein and CaO to the ultrasonic spray device, which generated small droplets of the mixture. The droplets fell from the tip of the ultrasonic nozzle in the methanol vapor atmosphere and then accumulated at the bottom, followed by circulation of the liquid with CaO to the ultrasonic spray device. This circulation was maintained until the desired reaction time. Methanol (Wako Pure Chemical) was vaporized and heated up to the desired temperature and then fed close to the bottom of the reactor. The methanol vapor flowed in the countercurrent direction of the falling droplets. The methanol vapor from the reactor was recovered by the condenser.

The experiments were conducted for 60 min, and the measurements were taken every 15 min. The experimental parameters investigated were triolein flow rates of 2.5, 6, and 9 mL/min, reaction temperatures of 70, 80, 90, and 100 °C, and catalyst loadings of 1, 3, 5, and 7 wt%, whereas methanol injection was kept constant at a liquid feed rate of 4.5 mL/min, which approximately corresponds to a volumetric flow rate of 3.08 L/min for the methanol vapor at the boiling point. A comparative experiment of the conventional transesterification in the liquid phase at 60 °C (below methanol's boiling point) was carried out by setting up the conditions, using a methanol to triolein ratio of 6:1, 5 wt% CaO, and mixing with the magnetic stirrer.

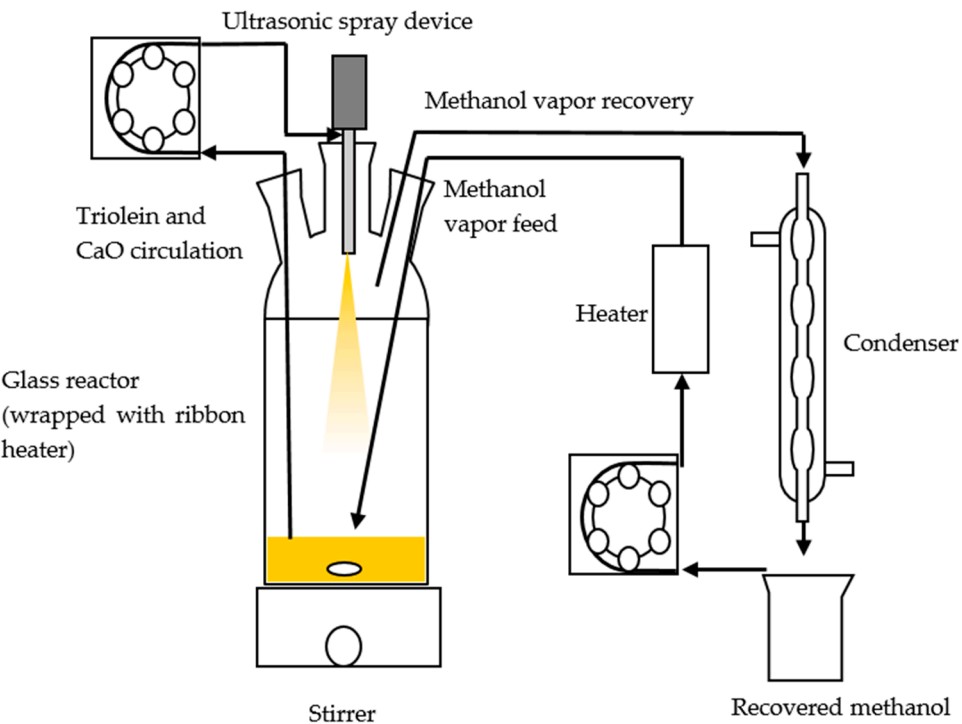

**Figure 1.** Schematic of the experimental apparatus with the three-phase reactor.

*2.3. FAME (Fatty Acid Methyl Ester) Analysis*

The yield of methyl ester was determined every 15 min by sampling approximately 2 mL of the product. The catalyst was removed from the sample using a syringe with a microfilter. As a standard, about 600 μL each of the sample and methyl heptadecanoate (Wako Pure Chemical) were prepared. Both the sample and the standard were weighed and mixed, and they were then diluted by 1.5 mL of *n*-heptane (Wako Pure Chemical) followed by analysis of the mixture. The methyl ester content in the mixture was calculated from the peak area obtained by a GC (Gas Chromatography) (Shimadzu GC-14B) equipped with a capillary column (DB-WAX 60 m) and a flame ionization detector. The carrier gas was nitrogen. The temperature of the injector and the detector were both set to 250 °C. The column temperature was initially maintained at 180 °C for 2 min and then increased to 230 °C at a heating rate of 10 °C/min. The complete analysis was conducted for about 9 min. The methyl ester yield was calculated in the same manner as Mun's calculation [10].

**3. Results and Discussion**

*3.1. Effect of the Flow Rate of Triolein on Methyl Ester*

Figure 2 shows the effect of the triolein flow rate on the methyl ester yield at 90 °C and 5 wt% CaO. The yield gradually increased with time. It was observed that a flow rate of 6.0 mm/min gave the highest yield, compared with those at other flow rates during the experimental time.

Figure 3a–c shows the pictures of the droplets using an optical microscope at triolein flow rates of 2.5, 6.0, and 9.0 mL/min, respectively. Table 1 shows the number and average diameter of the droplets by measuring the sizes of the droplets shown in Figure 3. The average diameter increased with an increasing flow rate, while the higher number of droplets was generated at 6.0 mL/min, compared to that at 9.0 mL/min. The total surface area of the droplets, which were estimated on the basis of the average diameter of the droplets and the flow rate, indicates that the highest surface area was obtained at a flow rate of 6.0 mL/min, causing the highest yield, as shown in Figure 2.

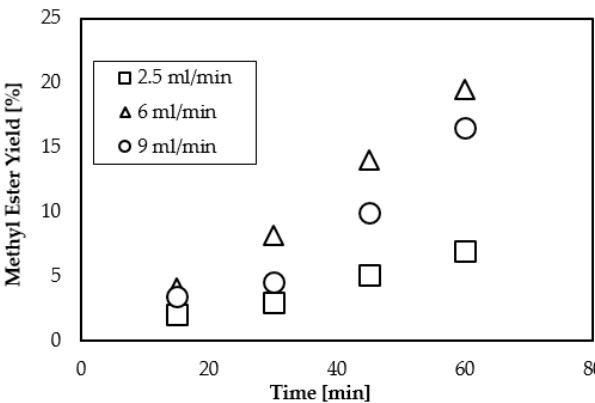

**Figure 2.** Effect of the triolein flow rate on the methyl ester yield at 90 °C and 5 wt% CaO.

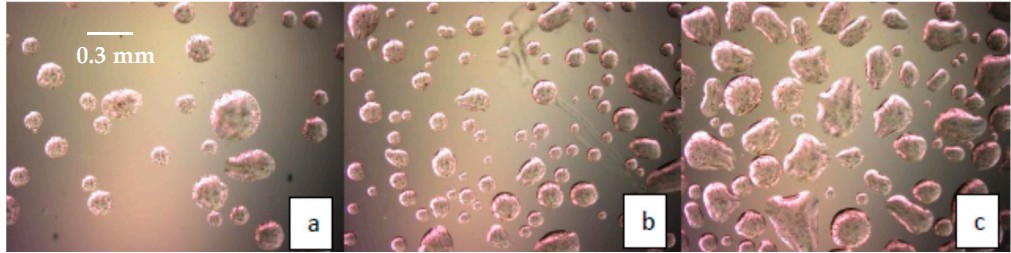

**Figure 3.** Pictures of droplets at triolein flow rates of (**a**) 2.5 mL/min, (**b**) 6 mL/min, and (**c**) 9 mL/min.

**Table 1.** Average diameters and surface areas of sprayed droplets.

| Triolein Flow Rate (mL/min) | Average Diameter of Droplets (mm) | Volume of One Droplet (mL) | Number of Droplets Sprayed (1/min) | Surface Area of One Droplet (cm²) | Total Surface Area per min (cm²/min) |
|---|---|---|---|---|---|
| 2.5 | 0.20 | $4.2 \times 10^{-6}$ | $6.0 \times 10^{6}$ | $1.2 \times 10^{-3}$ | $7.5 \times 10^{2}$ |
| 6.0 | 0.20 | $4.3 \times 10^{-6}$ | $1.4 \times 10^{7}$ | $1.3 \times 10^{-3}$ | $1.8 \times 10^{3}$ |
| 9.0 | 0.35 | $2.3 \times 10^{-5}$ | $4.0 \times 10^{6}$ | $3.5 \times 10^{-3}$ | $1.5 \times 10^{3}$ |

### 3.2. Effect of CaO Loading on Methyl Ester Yield

Catalyst loading is a principal factor for determining the transesterification process. To investigate the effect of CaO loading on the methyl ester yield, the experiments were carried out at four different catalyst loadings of 1, 3, 5, and 7 wt% at a triolein flow rate of 6.0 mL/min and a temperature of 90 °C. The distribution of CaO particles (Figure 4) increased with the increase of CaO loading. As shown in Figure 5, the yields did not significantly change between the four catalyst loadings in the beginning of the reaction and not until 30 min passed, depicting a trend that higher CaO loading results in a higher yield. The effect of CaO loading is a little pronounced from 45–60 min.

In particular, the yield proportionally increased from 10.0% to 14.1% at 45 min and 14.5% to 19.5% at 60 min, respectively, with increased CaO loading from 1 wt% to 5 wt%. The distribution of the higher CaO loading in the droplets improved the contact between the triolein and CaO, resulting in a higher yield of methyl ester. Similar results were reported by Stamenkovic [11] and Sarve [12]. Concerning the CaO loadings of 5 and 7 wt%, there were no significant yield differences during the first 45 min of the reaction. Moreover, the yields of the 5 and 7 wt% CaO loadings were 19.5% and 17.8%, respectively, at 60 min. This might have been caused by the reduction of the droplet diameter. It was observed that the average diameter of the droplet at a 5 wt% CaO loading was smaller than that at 7 wt%. After calculating the total surface area, it was found that a CaO loading of 5 wt% gave a

higher surface area than that at 7 wt%, and hence a higher methyl ester yield was obtained at a CaO loading of 5 wt%.

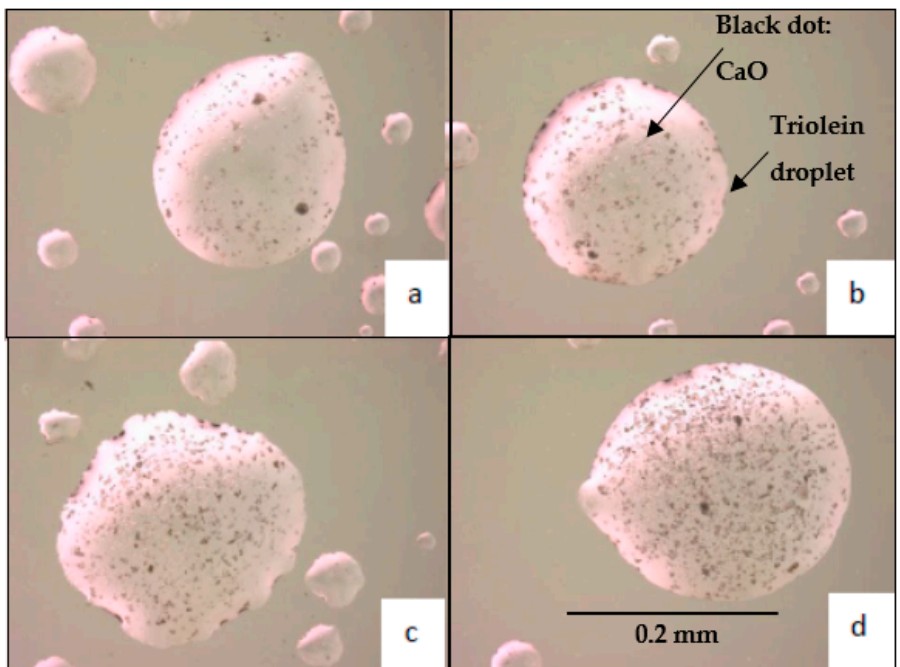

**Figure 4.** Pictures of droplets at CaO loadings of (**a**) 1 wt%, (**b**) 3 wt%, (**c**) 5 wt%, and (**d**) 7 wt%.

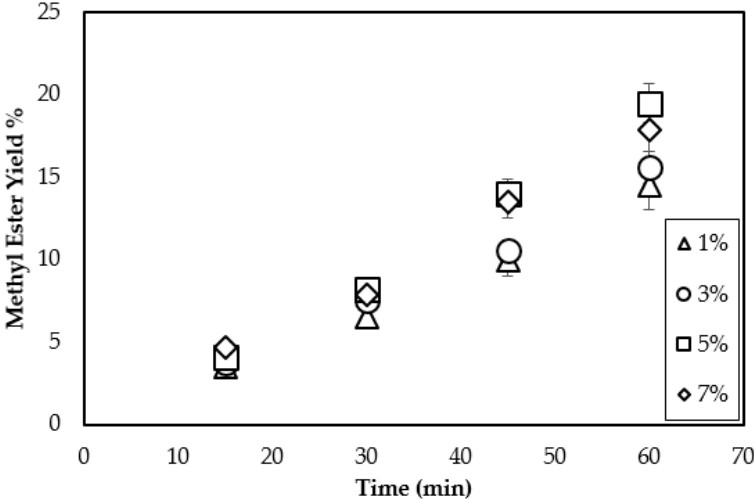

**Figure 5.** Effect of the catalyst loading on the methyl ester yield (90 °C and a triolein flow rate of 6.0 mL/min).

### 3.3. Effect of Temperature on the Methyl Ester Yield

Figure 6 shows the effect of the reaction temperature on the methyl ester yield at a triolein flow rate of 6.0 mL/min and 5 wt% CaO. It was observed that the yield increased with the reaction temperature and was higher with a longer reaction time. At 60 min, the yield increased from 12.5% to 23.8% as the temperature increased from 70 °C to 100 °C, because the increase in temperature accelerated the transesterification rate in the endothermic reaction [13], although an increase in the operating temperature resulted in a decrease in the solubility of the methanol [14].

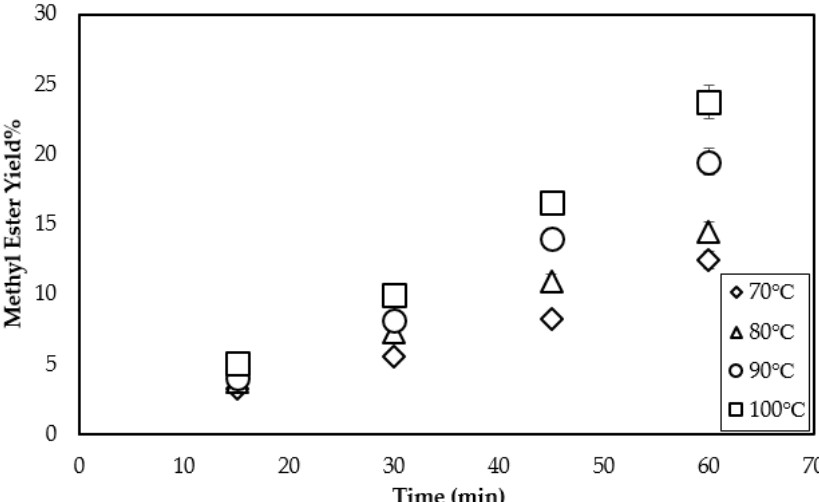

**Figure 6.** Effect of reaction temperature on the methyl ester yield (triolein flow rate of 6.0 mL/min and 5 wt% CaO).

*3.4. Comparison between Three-Phase with Ultrasonic Spraying and a Conventional Reactor*

The reaction time was extended to 120 min in order to find the behaviors of the reaction with the three-phase reactor using ultrasonic spraying at 90 °C. Additionally, the liquid phase reaction was carried out at 60 °C in a conventional reactor using a methanol to triolein ratio of 6:1. Figure 7 shows that the methyl ester yield of the three-phase reactor using ultrasonic spraying was 2–10% higher than the conventional batch reactor during 60 min of reaction. However, after 60 min, the yield still gradually increased with time in the three-phase reactor, whereas the yield suddenly increased after 60 min in the conventional reactor.

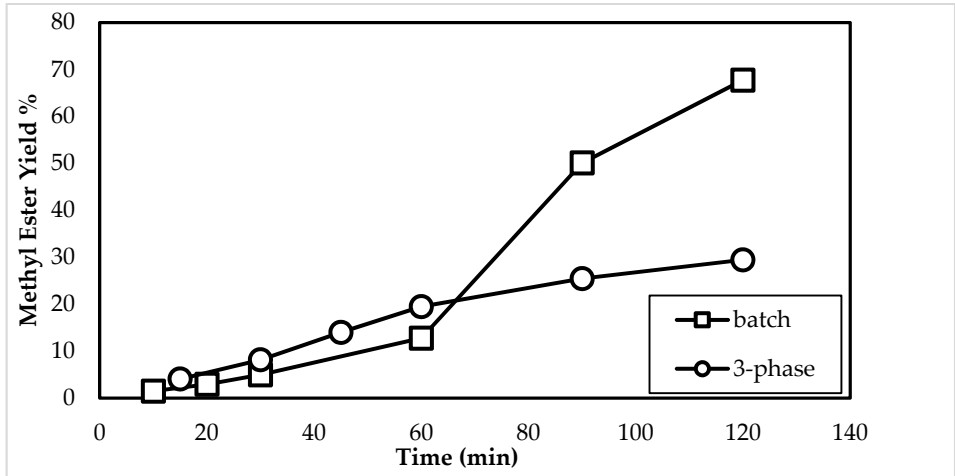

**Figure 7.** Comparison of methyl ester yields in the three-phase (90 °C) and conventional (60 °C) reactors.

Generally, triolein and methanol are not miscible and form two liquid phases at the initial stage of transesterification in the conventional reactor. Therefore, the mixing process is an important factor for enhancing mass transfer among two phases and heterogeneous catalysts, resulting in an accelerating reaction, as reported by Sarve et al. [14], Noureddini and Zhu [15], and Hingu et al. [16]. However, as methyl ester is formed, it acts as a mutual solvent for triolein and methanol, and there is a single phase where the transesterification is accelerated. This phenomenon corresponds to the sudden increase in the yield for the conventional reactor. On the other hand, as explained later, methanol mass transfer controls the transesterification, owing to the low solubility of methanol, causing a gradual increase

in the yield for the three-phase reactor even above a yield of 20%. From these results, it is concluded that the three-phase reactor is especially effective for the initial stage of transesterification. As a result, the transesterification process is preferable to be carried out using the three-phase reactor to achieve a certain yield, followed by the reaction in the batch reactor to reduce the reaction time with desired yield.

### 3.5. Relationship between the Time-Totalized Surface Area of the Droplets and the Methyl Ester Yield

The experimental results from Section 3.1 showed that the methyl ester yield depended on the triolein flow rate, as well as the surface area of the droplets generated per time. Thus, a quantitative analysis from Table 1 was used to calculate the time-totalized surface area (total surface area of the droplets produced until a certain reaction time). The area was considered to be the total area of contact between the methanol vapor and the triolein droplets during the experiment. Figure 8 shows the relation between the methyl ester yield and the time-totalized surface area for all triolein flow rate conditions. The yield behaves in one curve against the time-totalized surface area, suggesting that the time-totalized surface area was a key parameter in the three-phase reactor. It was found that the highest yield was with 19.5 wt% at the highest time-totalized surface area and a triolein flow rate of $10.8 \times 10^4$ cm$^2$ at 6 mL/min.

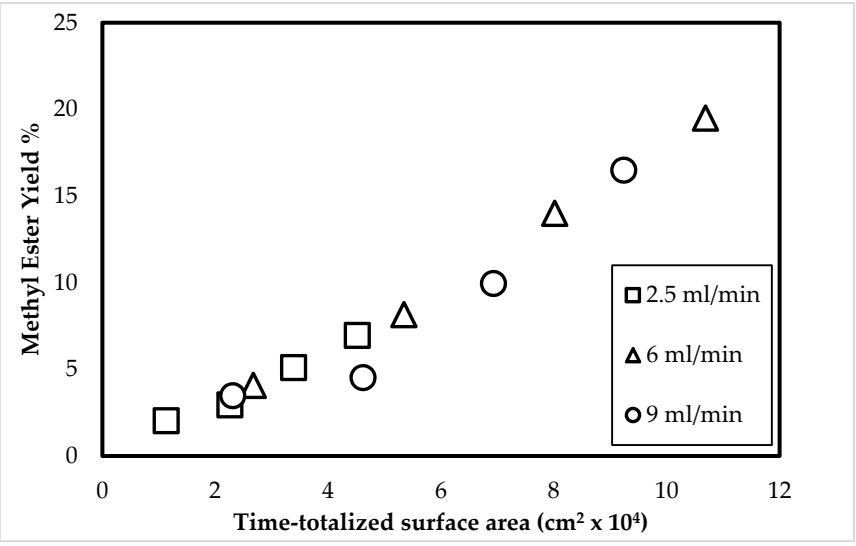

**Figure 8.** Effect of the time totalized surface area of the droplets on the methyl ester yield.

In the three-phase reactor, the transesterification took place using the following three steps:

- Step 1: Methanol mass transfer happens from the vapor phase to the triolein liquid droplets;
- Step 2: The dissolved methanol diffuses and reaches the catalysts in droplets;
- Step 3: The methanol reacts with the triolein in the presence of the catalysts.

Usually, Step 3 determines the overall transesterification rate because the intrinsic reaction rate is low. Owing to the low solubility of methanol to triolein, Step 1 became the rate-determining step, and the high reaction temperature enhanced the rate of Step 3. Therefore, the yield was strongly related to the time-totalized surface area, which directly affected the mass transfer of methanol.

## 4. Determination of the Kinetic Parameters

To evaluate the effectiveness of the three-phase reactor, the overall reaction rate of the transesterification was estimated, assuming that the reaction followed pseudo first-order kinetics, as proposed by Veljkovic et al. [17]:

$$Triolein + 3\ MeOH \rightleftarrows 3\ Methylester + Glycerol$$

The first-order rate equation is written as

$$-r_A = -\frac{dC_T}{dt} = kC_T = k\left(C_{T0} - \frac{1}{3}C_E\right) \tag{1}$$

where $C_{t0}$ is the initial concentration of triolein, $C_t$ is the concentration of triolein at time $t = C_{T0} - 1/3\ C_E$, and $C_E$ is the concentration of methyl ester at time $t$.

By integrating Equation (1), the solution is as follows:

$$-ln\frac{C_{T0} - \frac{C_E}{3}}{C_{T0}} = kt \tag{2}$$

Figure 9 shows the relation between the data $\left(-ln\frac{C_{T0} - \frac{C_E}{3}}{C_{T0}}\right)$ and $t$, where the slope of the linear line is $k$ (at a constant temperature) for the series of experiments conducted at different temperatures of 70, 80, 90, and 100 °C at a triolein flow rate of 6 mL/min and 5 wt% CaO. The results of $k$ are shown in Table 2, where the coefficient of determination $R^2$ nearly indicated unity, suggesting that the assumption of first-order kinetics is acceptable. The results indicated that a higher temperature enhanced $k$, as expected.

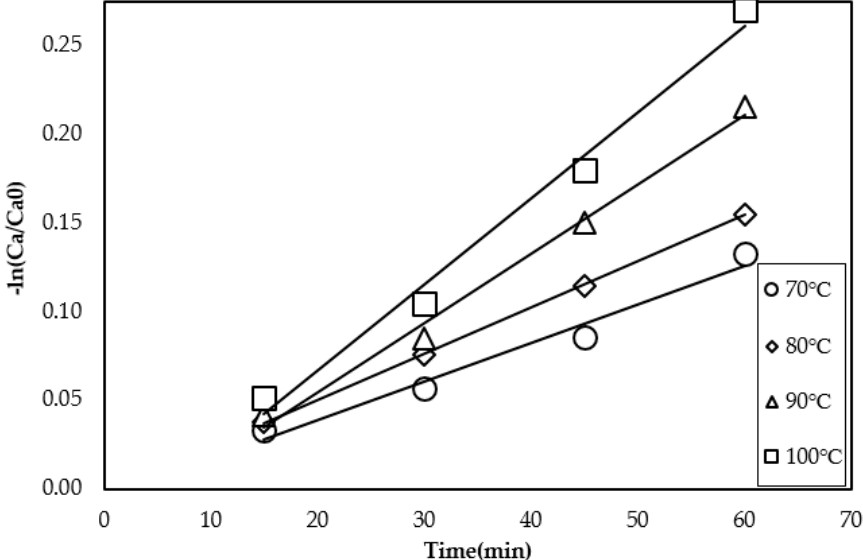

**Figure 9.** Concentration ratio of triolein versus the reaction time at different reaction temperatures and at a triolein flow rate of 6 mL/min and 5 wt% CaO.

**Table 2.** First-order rate constants for triolein transesterification with methanol on CaO. (Triolein flow rate of 6 mL/min and 5 wt% CaO.)

| Temperature | 70 °C | 80 °C | 90 °C | 100 °C |
|---|---|---|---|---|
| $k$, min$^{-1}$ | $1.21 \times 10^{-2}$ | $1.75 \times 10^{-2}$ | $2.63 \times 10^{-2}$ | $3.27 \times 10^{-2}$ |
| $r^2$ | 0.99 | 0.99 | 0.99 | 0.97 |

The activation energy of the reaction was estimated using the following Arrhenius Equation (3) with the data of *k*:

$$k = k_0 \, e^{\frac{-E}{RT}}$$
$$ln \, k = ln \, k_0 - \frac{E}{RT} \tag{3}$$

where $k_0$ is the pre-exponential factor (min$^{-1}$), *Ea* is the apparent activation energy (kJmol$^{-1}$), *R* is the gas constant (8.314 kJmol$^{-1}$K$^{-1}$), and *T* is the reaction temperature (K). As shown in Figure 10, the simple linear regression between *ln k* and $\frac{1}{T}$ was found to be a straight line, from which k0 and *Ea* were estimated as 64.9 min$^{-1}$ and 36.1 kJ mol$^{-1}$, respectively. Table 3 summarizes the range of *k* and *Ea*, calculated with the results of another similar piece of research. Vujicic [18] studied the kinetics of the production of sunflower biodiesel as a two-phase reaction with CaO. Their work had a higher activation energy (101.0 kJ mol$^{-1}$) because the composition of sunflower oil has several types of fatty acids, resulting in harder transesterification than triolein, which has only oleic acid. The two-phase study was also investigated by Anilkumar [19], using waste cooking oil as a raw material and CaO from eggshells prepared by calcination at 800 °C, followed by hydration at 60 °C, and then calcination at 600 °C again as catalysts. It should also be noted that the value for *Ea* (54.1 kJ mol$^{-1}$) was higher than this study because several triglyceride components were present in the oil. In the case of an ultrasound-assisted homogenous reaction, Parkar [20] observed a lower *k* value and a higher *Ea* value as compared to the present study. The rate constant of this work (1.21–3.70 × 10$^{-2}$ min$^{-1}$) was a little higher than other works using CaO, owing to the larger contact area and higher temperature. However, *k* was not enhanced as much under these conditions because the low solubility of methanol in triolein reduced the mass transfer of methanol.

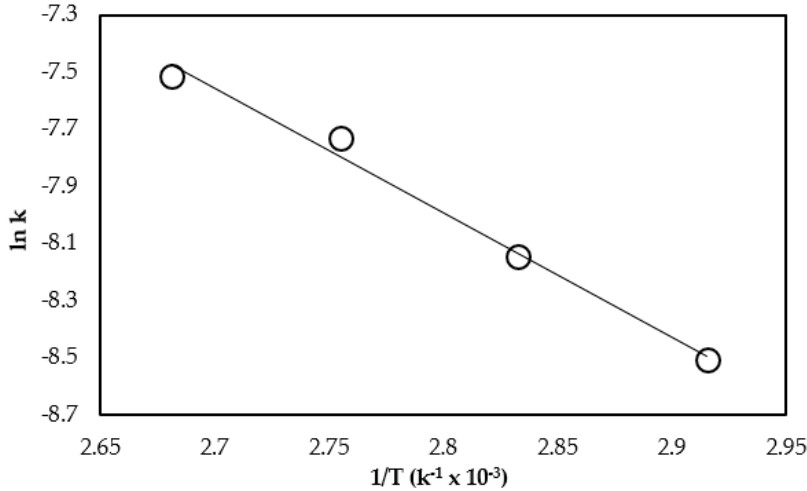

**Figure 10.** Arrhenius plots between ln *k* versus 1/T.

**Table 3.** Rate constants and activation energies of the reaction for any catalyst and type of transesterification.

| Raw Material or Type of Transesterification | Catalyst | Rate Constant *k* (min$^{-1}$) | Activation Energy *Ea* (kJmol$^{-1}$) | Reference |
|---|---|---|---|---|
| Triolein | CaO | 1.21–3.70 × 10$^{-2}$ | 36.1 | Present work |
| Sunflower | CaO | 0.27–5.92 × 10$^{-2}$ | 101.0 | [18] |
| Waste cooking oil | CaO from eggshells | 0.92–2.25 × 10$^{-2}$ | 54.1 | [19] |
| Soybean oil (ultrasound-assisted) | NaOH (homogenous) | 0.37–5.2 × 10$^{-3}$ | 55.4 | [20] |

## 5. Conclusions

The transesterification of triolein with methanol and CaO in a three-phase reactor was investigated. It was found that the optimum triolein flow rate producing the highest yield of methyl ester was mainly a result of the highest contact area of the triolein droplets generated by an ultrasonic spray. The three-phase reactor produced a yield 2–5% higher than the conventional reactor during a 60 min period, which confirmed the advantage of the three-phase reactor using an ultrasonic spray, in which the time-totalized surface area was the key factor to determine the yield. Comparing the temporal behavior of the yield in the conventional reactor, the combination of the three-phase reactor for the initial stage of the reaction and the batch reactor for the successive stage would be effective for fast transesterification.

**Author Contributions:** Conceptualization, R.V.; S.K. and H.S.; methodology, R.V.; writing—original draft preparation, R.V.; writing—review and editing, R.V.; S.K. and H.S. All authors have read and agreed to the published version of the manuscript.

**Funding:** This research received no external funding.

**Data Availability Statement:** Data is contained within the article.

**Conflicts of Interest:** The authors declare no conflict of interest.

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
