# Peer review of "Transesterification Using Ultrasonic Spray of Triolein Containing CaO Particles into Methanol Vapor in a 3-Phase Reactor"

_processes, doi:10.3390/pr9010181_

Round 1
Reviewer 1 Report
Dear authors
Congratulations for your work and your article. It was clear and well detailed. I don't have much comments to do.
Just a request, in the Figure 3 it would be possible to put the scale on the images? It will help to visualize the sizes.
Another comment, more directed to future works, when you study the influence of different parameters like temperature, flow rates , etc... you ever try to do a statistical analysis starting with an experimental design? instead of fix parameters who looks better, try to do a experimental matrix of the different parameters and do a statistical analyses of the influence of all at the same time and choose the best conditions in the end.
good work, keep going
Author Response
To Reviewer 1
Please see the attachment.

Reviewer 2 Report
The article is focusing on a three-phase transesterification of triolein. There is extensive experimental work. However, the discussion and the display of the results needs significant improvement before publication. My comments are the following:
Page 1 The title is confusing. CaO particles in methanol vapour?
Page 2 Line 68. Material and methods What was the volume of the reactor?
Page 3 Line 93 How the methanol fed into the system? It is not clear.
Page 4 Figure 2 Please explain this phenomenon, why the middle flow rate is better. What is the residence time in this case, or what is the optimal bubble size?
About Table 1, the number of droplets sprayed has a maximum at 6.0 ml/min, and the volume of one droplet at 9.0 ml/min, should not be both variables increase in one direction? Please explain.
Page 5 Figure 4 I presume the CaO is the white and the droplets are the dark parts. Please discuss it in detail.
Page 6 Figure 6 The axis, and the label overlaps. Please check.
Page 7 Figure 7 Is there any energetic gain? Because as I see the operation of the batch reactor is easier, there is no need for expensive equipment, just add the reagents, and remove the mixture after two hours. What is the importance of using this new method?
Page 8 Line 230 The authors have two equilibrium reaction here, and one reaction rate. Which reactions were considered here, and why as an irreversible.
Page 9 Figure 9 The axis, and the label overlaps. Please change. The y axis label is not the same than the caption.
Page 10 Figure 10 The axis, and the label overlaps. Please change.
Page 11 Line 299 The novelty should be more clear.
How can the two reactor types be combined?
Author Response
To reviewer 2
Please see the attachment.

Round 2
Reviewer 2 Report
The authors addressed all my concerns, therefore the paper can be accepted.